# Internet-Based Psycho-Physical Exercise Intervention Program in Mild-to-Moderate Depression: The Study Protocol of the SONRIE Randomized Controlled Trial

**DOI:** 10.3390/ijerph22040540

**Published:** 2025-04-01

**Authors:** Juan Manuel Escudier-Vázquez, Manuel Ruiz-Muñoz, Inmaculada Garrido-Palomino, Sonia Ortega-Gómez, Eulalio Juan Valmisa Gómez de Lara, María del Mar Espinosa Nogales, Alicia Viglerio Montero, Miguel Ángel Rosety-Rodríguez, David Jiménez-Pavón, Ana Carbonell-Baeza, Vanesa España-Romero

**Affiliations:** 1MOVE-IT Research Group, Department of Physical Education, Faculty of Education Sciences, University of Cadiz, 11519 Cadiz, Spain; juanmanuel.escudier@uca.es (J.M.E.-V.); manuelruiz94@icloud.com (M.R.-M.); inmaculada.garrido@gm.uca.es (I.G.-P.); sonia.ortega@uca.es (S.O.-G.); miguelangel.rosety@uca.es (M.Á.R.-R.); david.jimenez@uca.es (D.J.-P.); ana.carbonell@uca.es (A.C.-B.); 2Instituto de Investigación e Innovación Biomédica de Cádiz (INiBICA), 11009 Cádiz, Spain; 3C-HIPPER Climbing Research Association, Cadiz 11100, Spain; 4Department of Psychology, Sociology and Philosophy, Faculty of Education, University of Leon, 24071 Leon, Spain; 5Mental Health Service, Puerto Real University Hospital, 11510 Cadiz, Spain; eulalioj.valmisa.sspa@juntadeandalucia.es (E.J.V.G.d.L.); mmarespinosan@gmail.com (M.d.M.E.N.); viglerio@gmail.com (A.V.M.); 6CIBER of Frailty and Healthy Aging (CIBERFES), Instituto de Salud Carlos III, 28029 Madrid, Spain

**Keywords:** depression, mental health, internet-based cognitive–behavioral therapy, physical exercise, endocannabinoid system, randomized controlled trial

## Abstract

The COVID-19 pandemic has intensified depression due to isolation and reduced physical activity, highlighting the need for accessible remote treatments. The SONRIE study evaluates the effectiveness of a 12-week online intervention combining physical exercise and internet-based cognitive–behavioral therapy on depressive symptoms in adults with mild-to-moderate depression. This randomized controlled trial involved 80 adults aged 25–65 years diagnosed with depression according to the ICD-10 criteria. Participants were randomized to an experimental group receiving the combined online intervention or to a control group receiving standard care. The primary outcome was the change in depression severity, assessed by the Beck Depression Inventory, with outcomes measured at baseline, immediately post-intervention, and after an 8-week follow-up phase. Statistical analyses include analysis of covariance to compare group changes over time, with effect sizes quantifying the intervention’s impact. The SONRIE study demonstrates a promising online approach for treating depression, with potential implications for clinical practice and public health strategies.

## 1. Introduction

The COVID-19 pandemic has intensified global mental health challenges, primarily due to the isolating effects of confinement [1]. In Spain, stringent measures implemented on 14 March 2020 [2] effectively reduced virus transmission but also curtailed personal freedoms [2,3]. This led to a significant decrease in physical activity (PA) and heightened social isolation [4]. Such conditions have precipitated and exacerbated mental health disorders [5], including depressive symptoms [6], anxiety [5], and sleep disturbances [7]. It affects approximately 270 million individuals globally [8] and is projected to become the leading cause of disability by 2030 [6,9]. The urgency for effective interventions is clear. Consequently, the pandemic underscored the urgent need for accessible and effective remote interventions to mitigate these adverse mental health outcomes.

Recent randomized controlled trials (RCTs) have demonstrated that combined psychological and physical activity interventions are particularly effective for improving mental health outcomes in individuals with depression. For instance, Norouzi et al. (2024) found significant improvements in depressive symptoms, anxiety, perceived stress, and sleep quality following combined mindfulness-based stress reduction and physical activity interventions among patients with major depressive disorder [7]. Similarly, a systematic review by Khazaie et al. (2023) concluded that physical-activity-based interventions significantly enhance sleep quality in patients diagnosed with depression, highlighting exercise as a promising adjunctive treatment option [10]. Additionally, Rezaie et al. (2023) indicated that higher physical activity levels and better sleep quality significantly predict lower emotion dysregulation, a key aspect linked to depression [11]. Together, these findings strongly support the rationale for combining psychological and physical interventions to comprehensively target depressive symptoms and their associated factors.

Recent evidence suggests that the endocannabinoid system (ES) plays a critical role in modulating depressive symptoms [12], emotional processing [13,14,15], and neuroplasticity, particularly through its influence on brain-derived neurotrophic factor (BDNF) [12,16]. Dysfunctions in ES signaling, due to genetic factors or pharmacological disruptions, can contribute to depressive-like symptoms [17,18]. Notably, depressed individuals often exhibit lower plasma levels of endocannabinoids (eCBs) and their analogs, supporting the hypothesis of an endocannabinoid deficiency in depression [19,20]. Physical exercise may help counteract this deficiency by increasing circulating eCBs [21,22], especially anandamide (AEA) and 2-arachidonoylglycerol (2-AG), which act predominantly through CB1 receptors in brain regions and CB2 receptors in immune tissues [18]. Through these interactions, eCBs help regulate mood and stress response [12,16,23]. Despite these promising associations, existing research on the exercise-induced modulation of ES biomarkers in depressive disorders remains largely pre-clinical or observational, emphasizing the need for well-designed clinical studies [24,25]. Therefore, exploring the interplay between physical exercise, ES modulation, and BDNF signaling could offer valuable insights into novel therapeutic approaches.

Furthermore, internet-based cognitive–behavioral therapy (iCBT) has demonstrated efficacy in reducing depressive symptoms, improving emotional regulation, and facilitating cognitive flexibility in individuals unable or unwilling to engage in face-to-face interventions [26,27] through digital methods [28,29]. The internet-based delivery of CBT can significantly enhance treatment accessibility and adherence, addressing common barriers such as geographic limitations or reluctance to participate in face-to-face therapy [30,31]. The combination of physical exercise and iCBT, therefore, holds significant promise, potentially offering synergistic benefits by simultaneously addressing the biological, psychological, and behavioral dimensions of depression [32]. Prior studies have successfully combined physical activity with psychological interventions such as mindfulness, demonstrating robust improvements in emotional regulation and overall psychological health [7,10,11,33,34,35]. Nevertheless, the effectiveness and biological underpinnings of combining iCBT and physical exercise interventions delivered entirely online remain unexplored.

The SONRIE study (ClinicalTrials.gov Identifier: NCT05849792), originally launched in September 2019, is RCT designed to explore the effects of a 12-week online intervention combining iCBT and structured physical exercise in adults diagnosed with mild-to-moderate depression. Although initiated before the global outbreak of COVID-19, subsequent confinement periods have underscored the study’s relevance and applicability in scenarios of isolation. The primary aim of the SONRIE study was to evaluate the immediate and sustained effects of this combined online intervention on depressive symptoms. Additionally, the study has defined secondary objectives: (i) to examine the effects of the intervention on physical health parameters such as body composition, physical fitness, and blood biomarkers including plasma levels of eCBs (AEA and 2-AG) and their analogs or BDNF levels; (ii) to explore its influence on psychological dimensions such as anxiety, stress, emotional states, interoceptive awareness, and overall well-being; (iii) to investigate interrelationships among lifestyle factors including physical activity, sedentary behavior, sleep patterns, and dietary habits, and their predictive value on depressive outcomes; and (iv) to assess the sustainability of the intervention effects during an 8-week follow-up period without further treatment. We hypothesize that participants in the combined intervention group will experience significantly greater reductions in depressive symptoms compared to participants receiving usual care. Secondary hypotheses propose that improvements in depressive symptoms may be associated with positive changes in plasma levels of eCBs and their analogs, and BDNF signaling, alongside improved physical fitness, sleep quality, and emotional regulation. By clearly delineating primary and secondary outcomes, the SONRIE study aims to provide robust evidence on the clinical efficacy and biological underpinnings of an integrated, internet-based psycho-physical exercise intervention program for depression.

## 2. Materials and Methods

### 2.1. Study Design and Protocol Registration

The SONRIE study is an RCT registered at ClinicalTrials.gov (Identifier: NCT05849792). It was conducted within the Mental Health Clinical Management Unit at Puerto Real University Hospital (Cadiz). The study design was approved by the Research Ethics Committee (Reference: 1875-N-18). All participants provided written informed consent prior to participation, in accordance with the Declaration of Helsinki guidelines [36]. They were informed about the study’s purpose, procedures, potential risks (e.g., minor discomfort from blood sampling), potential benefits, and the confidentiality of their personal information. Participants were also advised of their right to withdraw from the study at any time without repercussion. Data were coded to ensure anonymity and stored securely with restricted access limited to the research team. Furthermore, liability insurance covered any potential damages related to study participation. Recruitment began in September 2019, and the study was completed in October 2020. The intervention was carried out between 9 March and 1 June 2020.

Initially planned as an in-person intervention, the SONRIE study protocol was adapted to a fully internet-based format due to the onset of the COVID-19 lockdown in Spain, coinciding exactly with the intervention start. This adaptation ensured intervention continuity and methodological rigor, providing consistent confinement conditions and adherence for all participants throughout the 12-week period.

### 2.2. Recruitment and Eligibility Criteria

Recruitment and diagnostic procedures were carried out at the Mental Health Unit of General Health Care at Puerto Real University Hospital. To engage the target population, outreach sessions were conducted, which included community presentations and social media campaigns. Interested individuals participated in informational meetings where they completed a pre-screening questionnaire to verify their eligibility according to the study criteria (Figure 1).

Eligible criteria included adults aged 25 to 65 years with a psychiatric diagnosis of mild-to-moderate depression according to the International Classification of Diseases 10th Revision (ICD-10) criteria [37]. All participants were required to be capable of engaging in PA without restrictions due to physical disability or health problems. Additionally, participants needed to have the ability to communicate effectively, to have the ability to read and understand the study’s main purpose, and to be willing and able to provide informed consent. Exclusion criteria included a diagnosis of major depression, the presence of acute or terminal illness, a history of cerebral infarction, epilepsy, or brain cancer, and unstable cardiovascular disease or other medical conditions that could interfere with participation in physical exercises.

### 2.3. Sample Size and Randomization

To detect a minimum mean difference of 8.26 in the primary outcome (depression symptoms), with an assumed standard deviation of 9.36, a sample size of 50 participants (25 per group) was calculated. These assumptions were based on the results of Stelzer et al. [38], who conducted an RCT involving individuals with depression undergoing a structured physical exercise psychotherapy intervention. Although their intervention modality was not delivered online, the target population and outcome measure were comparable to those used in the present study. The observed change in depressive symptoms in that trial was considered clinically meaningful and therefore appropriate to inform the sample size estimation in the context of exercise-based interventions for depression. This calculation assumed an alpha error of 0.05 and a statistical power of 85%. To accommodate potential loss to follow-up, 80 participants were initially recruited, accounting for a possible dropout rate. This ensures sufficient statistical power for the SONRIE study throughout its duration.

Participants were randomized in a 1:1 ratio to either the Experimental Group (EG; n = 40) or Control Group (CG; n = 40), ensuring balance in sex, age, and clinical severity of depression. The randomization process was conducted using a computer-generated sequence through STATA version 16.0 (Stata Corp, College Station, TX, USA). To prevent selection bias, allocation concealment was ensured by using a centralized randomization procedure, where the sequence was generated and stored by an independent researcher not involved in recruitment, intervention, or data collection. Group assignments were only revealed to participants after baseline assessments were completed, ensuring that neither researchers nor participants could predict or influence the allocation process.

After randomization, four participants from the CG withdrew, leaving 36 participants in this group. Subsequently, the impact of the COVID-19 pandemic led to further attrition, reducing the CG to 25 by the post-intervention assessment and to 16 by the 8-week follow-up. In contrast, the EG showed better retention, with 37 participants remaining at the end of the study. Of these, 30 attended over 80% of the sessions. A flowchart detailing participant progression is presented in Figure 1. The higher dropout rate observed in the CG may be partially explained by reduced motivation in the absence of an active intervention. In addition, elevated attrition is a well-documented phenomenon in digital health trials, especially in the context of self-guided or internet-based interventions. As described in the “law of attrition” [39,40], such patterns are often expected and reflect typical user behavior rather than methodological flaws. Similar adherence challenges have been reported in previous web-based trials for depression and anxiety [39,40].

### 2.4. Data Collection

The SONRIE study implemented a comprehensive clinical assessment protocol for both primary and secondary outcomes at three key time points: baseline (pre-test), immediately after the 12-week intervention period (post-test), and following an 8-week follow-up phase without further intervention (Figure 2). The complete clinical evaluation was structured into three visits, ensuring thorough data collection across all domains. All primary and secondary outcomes were assessed using standardized tools and validated questionnaires, as outlined in Table 1.

Given the nature of our intervention, complete blinding was not feasible because both participants and exercise facilitators were aware of group allocation. To minimize assessment bias, the assessors responsible for primary and secondary outcome measurements did not have access to participants’ previous results (baseline or prior follow-up) at the time of subsequent assessments (post-intervention and 8-week follow-up).

### 2.5. Primary Outcome

*Depressive symptoms* were assessed using the Beck Depression Scale (BDI), a widely utilized 21-ites scale that measures the severity of depression. This scale evaluates a range of symptoms including sadness, crying, loss of pleasure, feelings of failure and guilt, suicidal ideation, and pessimism. Scores range from 0 to 63, with higher scores indicating more severe depressive symptoms. The scoring is categorized as follows: 0–13 for minimal depression, 14–19 for mild depression, 20–28 for moderate depression, and 29–63 for severe depression. This scale allows for a detailed assessment of depressive symptoms intensity [41,42].

### 2.6. Secondary Outcomes

In addition to the primary outcome (BDI), a set of secondary outcomes was selected to provide a comprehensive assessment of the intervention’s effects on both physical and psychological health. The following measures were chosen based on their established relevance in depression research [6,9,11,12,43], as well as their proven validity, reliability, and sensitivity in capturing key domains that may be affected by the intervention.

#### 2.6.1. Physical Domain

*Body composition*. A multifrequency bioimpedance analyzer (TANITA-MC780MA) was used to measure fat mass (kg), fat mass percentage, lean mass (kg), and lean mass percentage [44]. Participants were asked to urinate before the assessment, and fast for at least 2 h prior to the assessment. Body mass index was calculated as weight (kg) divided by height squared (m^2^). Waist and hip circumferences were measured in cm with an anthropometric tape (Lufkin W606PM, Washington, USA), following standardized guidelines [45].

*Blood pressure and heart rate.* Resting blood pressure and heart rate were assessed using an Omron M6 upper arm blood pressure monitor (Omron Health Care Europe B.V., Hoofddorp, The Netherlands). Measurements were taken twice, spaced two minutes apart, following a 5 min period of rest. Participants were seated with the non-dominant arm positioned at heart level, the cuff placed on the upper arm, and the palm facing upwards to ensure accuracy.

*Objective PA levels and sedentary time* were evaluated using a triaxial accelerometer (Actigraph GT3X, MTI, USA; ActiLife Software version 6.13.3) [46]. Participants wore the device for seven consecutive days on the waist at the center of the lower back, ensuring 24 h recording except during water-based activities. A valid day was defined as at least eight hours (480 min) of wear time, and data were considered valid if the participant recorded at least three valid days, including at least one weekend day [46]. Non-wear time was identified as 60 consecutive minutes of zero counts, with allowance for a 2 min interval of non-zero values, and validated using Troiano cutoff points [47]. Data were collected over 60 s epochs at a sampling rate of 30 Hz [46]. To ensure comparability with previous research, Freedson’s cutoff points were applied to classify PA intensities [48]. Additionally, daily step counts and sedentary bouts lasting ≥30 min were analyzed to provide a more detailed understanding of activity patterns. In addition, self-reported PA was collected through the Global PA Questionnaire (GPAQ), which has been validated [49] and adapted to the Spanish context [50].

*Health-related physical fitness* was assessed through cardiorespiratory fitness (CRF), muscle strength, flexibility, agility, balance, and gait speed. CRF was assessed using the 6-Minute Walk Test, in which participants walked as far as possible on a 60 m circuit within six minutes [51]. Additionally, the Step Test was performed by stepping up and down a 30 cm platform at 96 beats per minute for 3 min. Heart rate was measured before, immediately after, and one minute post-test using a Polar H10 heart rate monitor (Polar Electro, Kempele, Finland) [52]. Muscle strength was assessed through targeted tests for both upper and lower limbs. Lower-limb strength was evaluated using three tests: the Chair Stand Test, which measured the number of sit-to-stand repetitions in 30 s [52,53]; the Five-Repetition Chair Stand Test, conducted once and recording the time to complete five repetitions as fast as possible [54]; and the Standing Long Jump Test, performed twice, with the best distance recorded [55,56]. Upper-limb strength was gauged using the Arm Curl Test, where participants completed maximum number of curls in 30 s, with weights of 2.3 kg for women and 3.6 kg for men [52,57]; and handgrip strength was measured with a digital dynamometer (TKK 5101 Grip-D, Tokyo, Japan). The test was performed twice per hand, with a 1 min rest between attempts [58], and grip span was adapted to hand size for men and women [59]. Flexibility was assessed using the Back Scratch Test for upper-body flexibility, which measured the distance in centimeters between the middle fingers, with one hand reaching over the shoulder and the other behind the back. Positive or negative values were recorded on both sides [52]; Lower-body flexibility was evaluated using the Chair Sit-and-Reach Test, in which participants, seated on a chair, extended one leg forward and reached toward their toes. The distance between fingers and toes was recorded for both legs [52]. Agility was assessed with the 8-Foot Up-and-Go Test, in which the best of two timed attempts to stand from a chair, navigate around a cone placed 8 feet away, and return to the chair was recorded [52]. Balance was assessed with the Unipedal Stance Test, where participants stood on one leg for up to one minute or until balance was lost. The test was performed twice per leg, and the best result for each was recorded [60]. Static balance was further tested in side-by-side, semi-tandem, and tandem positions, each held for 10 s [61]. Gait speed was assessed by gait speed using the 4 m and 6 m walk tests. Participants walked the marked distances at their usual pace, and time was recorded in seconds for each [61]. Finally, subjective fitness was assessed using the adapted Spanish version of the International Fitness Scale, which includes five response categories (very poor, poor, average, good, very good) [62].

*Blood biomarker analyses* were collected from each participant after an overnight fast (between 08:00 and 09:00 h) via venipuncture. The blood was drawn into K2-EDTA tubes and kept on ice until processing, which commenced within 10 min post-extraction to preserve sample integrity. Tubes were centrifuged at 1700× *g* for 15 min at 4 °C to separate plasma and serum. One milliliter (1 mL) of plasma was then aliquoted into cryotubes, and, for ES analyses specifically, 8 µL of orlistat (250 µg/mL in ethanol) were added to each aliquot to prevent analyte degradation. All samples were stored at –80 °C until further analysis. Where serum was required (e.g., BDNF measurements), blood was collected in serum-separator tubes and processed according to standard laboratory protocols. Biomarkers for the ES, neuroplasticity, and cardiovascular disease risk were measured. For the ES, plasma level concentrations of eCBs (AEA, 2-AG) and their analogs [oleoylethanolamide (OEA), linoleoyl glycerol (2-LG), oleoyl glycerol (2-OG), dihomo-γ-linolenoyl ethanolamide (DEA), docosahexaenoylethanolamide (DHEA), linoleoylethanolamide (LEA), and stearoylethanolamide (SEA)] were obtained following a previously validated method [63]. Briefly, 0.5 mL of thawed plasma was transferred to 12 mL glass tubes, spiked with deuterated internal standards (e.g., AEA-d4, DHEA-d4, LEA-d4, OEA-d4, 2-AG-d5, and 2-OG-d5), and extracted with tert-butyl methyl ether. After evaporation to dryness, the extract was reconstituted in 100 µL of water: acetonitrile (10:90, *v*/*v*) with 0.1% formic acid, then transferred to HPLC vials. Analyses were performed using liquid chromatography–tandem mass spectrometry (LC-MS/MS) [64,65] on a Waters Acquity UPLC system coupled to a Xevo TQ-S micro–Mass Spectrometer, with a BEH-C18 column (2.1 × 100 mm, 1.8 µm) at 55 °C and a flow rate of 0.4 mL/min. The mobile phase comprised 0.01% (*v*/*v*) formic acid in water (solvent A) and 0.01% (*v*/*v*) formic acid in acetonitrile (solvent B), and detection used multiple reaction monitoring (MRM). Quantification was achieved by isotope dilution, with quality control samples (injected across four batches) yielding RSDs of 7.0–10.7%. For the neuroplasticity assessment, BDNF was measure from serum using an indirect enzyme-linked immunosorbent assay (ELISA) utilizing a specific kit (RAyBio Human BDNF ELISA Kit, Chemicon, Temecula, CA, USA) designed for high sensitivity and specificity [66]. The assay was performed according to the manufacturer’s instructions, with each sample measured in duplicate, and results were expressed in ng/mL. Cardiovascular risk factors (total cholesterol, HDL-cholesterol, LDL-cholesterol, and triglycerides), glycemic control markers (glucose, insulin, glycated hemoglobin HbA1c, and insulin resistance assessed by HOMA-I), inflammatory markers (high-sensitivity C-reactive protein, tumor necrosis factor-alpha, and interleukin 6), and liver metabolism enzymes (alanine aminotransferase, aspartate aminotransferase, and alkaline phosphatase) were measured using standardized clinical protocols in accordance with the Andalusian Public Health System Biobank guidelines. Complete blood counts (hemoglobin, hematocrit, leukocytes, platelets), renal function (creatinine, urea), and electrolytes (sodium, potassium) were also measured using routine automated laboratory methods.

#### 2.6.2. Psychological Domain

*Anxiety and stress.* Symptoms were quantified using the Generalized Anxiety Disorder 7-item (GAD-7) scale, a validated tool that measures the severity of generalized anxiety across several dimensions including nervousness, worry, and restlessness. This scale rates symptoms from 0 (“not at all”) to 3 (“nearly every day”), with total scores ranging from 0 to 21 [67].

*Emotional states* were assessed using the validate Positive and Negative Affect Scale (PANAS) [68,69,70]. The scale consists of two subscales, each composed of 10 items. The Positive Affect subscale measures feelings associated with activation and pleasure, with higher scores (ranging from 10 to 50) indicating a greater positive affect. The Negative Affect subscale captures emotions linked to distress and discomfort, where higher scores indicate a greater negative affect. Participants rated the extent to which they had experienced each adjective over the past few weeks.

*Interoceptive awareness.* Interoceptive awareness was evaluated using the Multidimensional Assessment of Interoceptive Awareness (MAIA), which consists of 32 items spanning various dimensions of interoception. These dimensions include the ability to notice subtle bodily sensations, not being distracted or distressed by them, regulating attention to the body’s signals, and developing emotional awareness related to these sensations. Participants rated each item on a scale from 0 (never) to 5 (always), with higher scores indicating greater interoceptive awareness and better regulation of bodily signals [71].

*Well-being and quality of life.* Well-being was evaluated using two validated instruments. Ryff’s Psychological Well-Being Scale assesses key facets of psychological functioning such as self-acceptance and personal growth through 39 items. Total scores ranged from 39 to 234; higher scores indicate greater psychological well-being [72,73]. Additionally, health-related quality of life was assessed using the validated Spanish version of the 36-item Short-Form Health Survey (SF-36). The SF-36 measures eight dimensions: physical functioning, role limitations due to physical health, bodily pain, general health, vitality, social functioning, role limitations due to emotional problems, and mental health. Scores for each dimension range from 0 to 100, with higher scores indicating better health-related quality of life [74].

#### 2.6.3. Control Variables

Different variables are recorded that may influence the study results even though they are not part of the intervention: sex, age, educational level, marital status, occupational status, and income level; personal history of depression (depression duration) and medication use; information on the presence of comorbidities; sleep time and quality (accelerometer); adherence to a Mediterranean diet, measured using the PREDIMED questionnaire [75,76]; and usual intake estimates of food groups, energy, and nutrients using a self-administered semi-quantitative food-frequency questionnaire (FFQ) [77] and 5-day recalls [78].

### 2.7. Internet-Based Psycho-Physical Exercise Intervention Program

#### 2.7.1. Control Group (CG, Usual Care)

Participants randomly assigned to the CG received standard care throughout the study. This group did not engage in the internet-based psycho-physical exercise intervention program but continued their usual care routines.

#### 2.7.2. Experimental Group (EG, Enhanced Care with Physical Exercise and iCBT)

The psycho-physical exercise intervention implemented in this study refers to a structured and coordinated program that integrates a multicomponent physical exercise protocol with iCBT. Delivered entirely online, this dual approach was designed to address both the physiological and psychological dimensions of depression. The physical exercise component included balance, resistance, and CRF training, individually tailored and progressive, while the iCBT program was based on the Unified Protocol and targeted emotional regulation, cognitive restructuring, and behavioral activation. Both components were scheduled in a complementary manner, with participants viewing the psychological modules prior to each supervised exercise session, thereby enhancing the integrative therapeutic effect of the intervention.

The exercise protocol followed the Consensus on Exercise Reporting Template (CERT) for transparency and replicability [79]. Sessions were conducted by a Physical Educator and Sports Sciences Specialist registered according to Spanish law and holding a Master’s degree in Physical Activity and Health, with initial support provided by a physical therapist. This 12-week exercise program consisted of three weekly sessions: one supervised online session each Tuesday and two asynchronous unsupervised sessions. Sessions were delivered through private YouTube videos, accessible only to study participants, thereby maintaining confidentiality while providing on-demand exercise guidance. Exercise intensity was individually prescribed using the Borg Rate of Perceived Exertion (RPE) scale, targeting moderate to vigorous intensity as recommended by the American College of Sports Medicine’s (ACSM) [80]. Exercise volume adhered to the World Health Organization’s 2020 physical activity guidelines for adults [80].

Each 60–90 min session began with a 7–10 min warm-up involving dynamic stretches (e.g., arm circles, leg swings) and mobility exercises (e.g., hip rotations, ankle rolls) to gradually increase heart rate and prepare the body for physical activity. The main segment comprised balance, resistance, and CRF training. Balance training lasted 10 min per session, designed to improve coordination, postural control, and core stability. Participants progressed from static balance exercise to dynamic and advanced coordination tasks, maintaining a low perceived intensity (RPE 1–3). Resistance training occupied 25–35 min per session, emphasizing the progressive enhancement of muscular strength, motor control, and core stability. Exercises included squats, lateral and frontal lunges, biceps curls, push-ups, bent-over rows, and shoulder presses structured into 3–4 sets (8–12 repetitions or duration-based sets), following ACSM guidelines [81]. Progression involved increasing time under tension (TUT) principles (from week 4 onwards) and introducing light-to-moderate external weights [82]. The training intensity progressively increased from RPE 3–4 initially to RPE 7–8 by week 12. The CRF training, lasting 13–14 min, included aerobic exercises such as dance aerobics and step routines. Sessions were structured into 2–3 segments of 2 to 9 min each with brief rest intervals (45–60 s). The training intensity started at RPE 3–4 and gradually increased to RPE 7–8 by week 12, designed to achieve 55–89% of the participant’s heart rate reserve, consistent with ACSM recommendations [81]. The sessions concluded with an 8-10 min cool-down period incorporating stretching for major muscle groups, relaxation techniques, and postural adjustments. Participants completed pre- and post-session Google Forms questionnaires for supervised and unsupervised sessions to track attendance and monitor participant safety by collecting information on perceived exertion and satisfaction. The supervising specialist reviewed these responses regularly to detect signs of excessive fatigue, discomfort, injury, or dissatisfaction, and provided personalized support or adjustments to the program as needed. In addition, weekly adherence to the prescribed exercise sessions was monitored by analyzing these logs. Participants who missed sessions or reported difficulties received immediate individualized follow-up from the supervising fitness specialist to ensure ongoing engagement. Successful adherence was defined as completion of at least 80% of planned sessions. (Detailed session structure and progression are available in Table A1).

The psychological protocol of the intervention consisted of the iCBT program, adapted from a face-to-face Unified Protocol for the transdiagnostic treatment of emotional disorders [83]. It was conducted by a psychologist registered in accordance with Spanish law, holding a Master’s degree in Sports Psychology, a Master’s degree in Neuropsychology, and a specific qualification in the Unified Protocol. Ethical considerations were strictly adhered to throughout the intervention. The iCBT program comprised 18 modules addressing four main dimensions: physiological, emotional, behavioral, and cognitive aspects (Table A2). Modules included asynchronous, video-guided sessions lasting approximately 30 min each, delivered individually prior to the supervised exercise session. Following an introductory first session, participants received two sessions per week from weeks 2 to 7, and one session per week from weeks 8 to 12. Each module sequentially built upon the previous one, ensuring a cohesive and progressive learning experience. The content of the iCBT modules integrated psycho-educational components on cognitive skills (e.g., identifying and modifying negative thoughts and emotions), behavioral techniques (e.g., behavioral activation, posture modification), emotional regulation strategies (bottom-up and top-down approaches), coping skills (problem-solving and reducing avoidance behaviors), and insights into neurochemical regulation of mood (serotonin, dopamine, endorphins, and oxytocin). Participants received practical exercises and interactive tasks in each module, facilitating skill implementation and practice. Participants completed the PANAS before and after each supervised exercise session. As iCBT modules were scheduled to be viewed immediately before these sessions, PANAS scores also served to assess emotional responses to the psychological component of the intervention. These data, along with weekly feedback forms, were reviewed by the supervising psychologist to monitor adherence and detect signs of emotional distress or disengagement. Participants were encouraged to report any psychological discomfort, and the psychologist remained available throughout the program to provide support or intervention adjustments as needed. Successful adherence to the iCBT component was defined as completion of at least 80% of the modules.

### 2.8. Statistical Analysis

To assess the effectiveness of randomization, baseline (pre-test) values will be compared between the EG and the CG for potential clinically important differences using independent samples *t*-tests for continuous variables and chi-square tests for categorical variables. Effect sizes will be calculated to quantify the magnitude of any differences observed. The difference between groups in the change (pre, post, and 8-week follow-up) of the primary outcome (e.g., depression scores) and all secondary outcomes will be analyzed using Analysis of Covariance (ANCOVA) for continuous outcomes that meet model assumptions. For outcomes that are not continuous or do not meet ANCOVA assumptions, appropriate generalized linear models will be used, specifying the distribution and link function according to the nature of the data (e.g., logistic regression for binary outcomes). The change score will be the dependent variable, with group (EG or CG), time points (e.g., post-intervention, follow-up), and their interaction included as fixed effects. Baseline values and relevant confounding variables (e.g., age, sex) will be included as covariates to adjust for initial differences and control for potential confounders. Model assumptions (e.g., normality, homogeneity of variances, linearity) will be assessed using diagnostic plots and statistical tests. In cases where assumptions are violated, data transformations or non-parametric alternatives will be considered. Effect sizes (with 95% confidence intervals) and statistical significance will be reported for the effects of group (between-subjects), time (within-subject), and the group × time interaction. Partial eta squared (ηp^2^) will be reported for ANCOVA, with effect sizes interpreted as small (ηp^2^ ≥ 0.01), medium (ηp^2^ ≥ 0.06), and large (ηp^2^ ≥ 0.14) according to Cohen’s guidelines [84,85]. Statistical significance will be set at *p* < 0.05. To address the issue of multiple comparisons, *p*-values will be adjusted using the Bonferroni correction [86]. The primary analysis will follow a per-protocol approach, including only participants who attended at least 80% of the intervention sessions. Missing data will be handled using an intention-to-treat (ITT) approach, ensuring that all randomized participants are included in the analysis. A dropout analysis will be conducted to determine whether participants who withdraw differ significantly from those who complete the study. Additionally, sensitivity analyses will be performed to confirm the robustness of the results under various assumptions about missing data. As high attrition is common in internet-based interventions for depression [39,40], these strategies are essential to reduce the risk of attrition bias and preserve internal validity [87]. All statistical analyses will be performed using STATA version 16.0 (StataCorp LP, College Station, TX, USA).

## 3. Discussion

The SONRIE study presents an innovative approach to treating mild-to-moderate depression by integrating an online physical exercise program with iCBT. This integration, pioneering in its online format, addresses both the psychological and physiological aspects of depression, with the potential to overcome geographical and social barriers, that limit access to traditional therapy. The relevance of this study is heightened in the current context, where the COVID-19 pandemic and restrictions on face-to-face intervention [88] have driven the need to remote health care solutions. The SONRIE study, by utilizing online platforms for intervention delivery, aligns with this emerging need [31,32] and offers a viable alternative for individuals unable to access face-to-face therapy [89].

Integrating physical exercise with iCBT within an RCT framework represents a significant methodological contribution to the scientific literature. Unlike previous studies that have focused on combined physical exercise and CBT interventions in person [90,91], the online approach of our study facilitates greater flexibility and reach, potentially enhancing adherence and long-term outcomes. The study design enables the examination of the synergistic effects between physical and psychological interventions, providing a scalable model adaptable for diverse populations. Nevertheless, a challenge commonly reported in digital interventions is participant attrition, particularly in control groups not receiving active treatment [39,40]. Anticipating this, our analysis plan includes intention-to-treat and sensitivity approaches to address potential attrition bias and ensure the robustness of future findings [87]. Furthermore, the use of validated assessment tools ensures the generation of reliable and consistent data to evaluate the impact of the intervention on mental and physical health outcomes. The SONRIE study has the potential to serve as a model for future research and clinical practice, demonstrating how combined online interventions can effectively address depressive symptoms and expand the research and effectiveness of mental health services.

However, certain challenges are anticipated. The reliance on self-reported measures may introduce reporting biases [87], and the requirement for participants to have internet access and digital literacy may limit the generalizability of the findings to populations with limited technological resources. Future studies should consider strategies to include these populations, such as providing technological support or utilizing alternative delivery methods.

## 4. Conclusions

In conclusion, the SONRIE study represents a promising approach to treating mild-to-moderate depression through the online integration of physical exercise and iCBT. The study findings have the potential to inform clinical practice and guide future research exploring the underlying biological mechanisms and the adaptability of the intervention to diverse populations.

## Figures and Tables

**Figure 1 ijerph-22-00540-f001:**
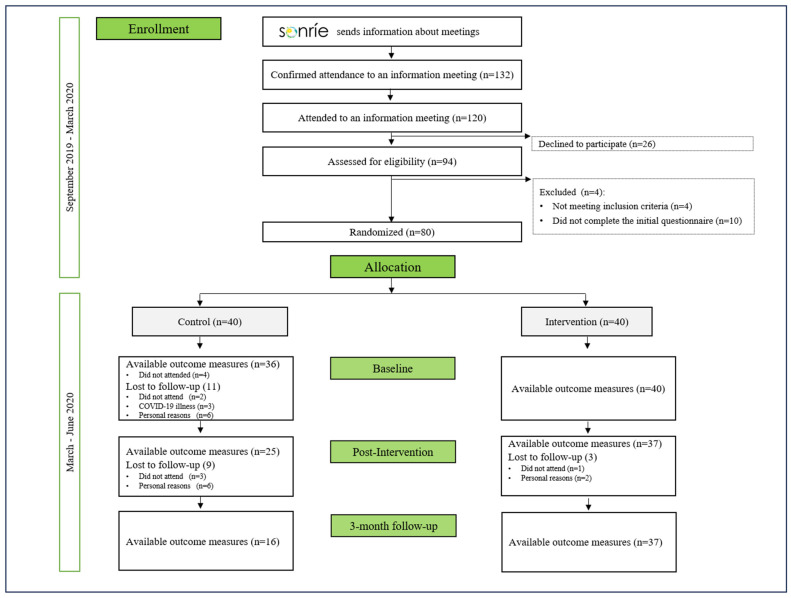
CONSORT flow diagram of the SONRIE Study.

**Figure 2 ijerph-22-00540-f002:**
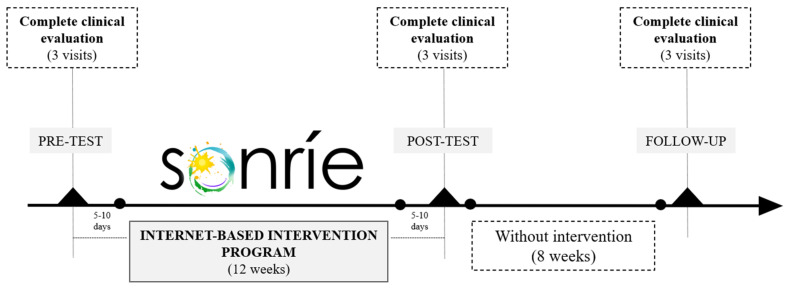
SONRIE data collection diagram.

**Table 1 ijerph-22-00540-t001:** Assessment tools for primary and secondary outcomes in the SONRIE study.

Outcome	Assessment Tool
**Primary outcome**	
Depressive Symptoms	- Beck Depression Inventory (BDI)
**Secondary outcome**	
**Physical Domain**	
Body Composition	- Weight, fat mass (kg), fat-free mass (kg), and body fat (%) by TANITA MC-780MA Bioimpedance Analyzer - Height (cm) by a stadiometer - Waist and hip circumferences by an anthropometric tape
Blood Pressure and Heart Rate	- Omron M6 Monitor upper arm blood pressure monitor
Physical Activity and Sedentary Time	- ActiGraph GT3X Accelerometer - Global PA Questionnaire (GPAQ)
Health-Related Physical Fitness	- CRF by the 6-Minute Walk test and the 3-Minute Step test - Muscle strength by the Chair Stand, the Standing Long Jump, the Arm Curl, and the Handgrip strength tests - Flexibility by the Back Scratch and Chair Sit and Reach tests - Agility by the 8-Foot Up-and-Go test - Balance by the Unipedal Stance test - Gait speed using the 4 m and 6 m walk tests - Self-reported fitness by International Fitness Scale
Blood Biomarker Analyses	- Endocannabinoid levels (AEA; 2-AG; OEA; 2-LG; 2-OG; DEA; DHEA; LEA; SEA) - Lipid and glycemic profiles, inflammatory markers and liver enzymes - Brain-Derived Neurotrophic Factor (BDNF)
**Psychological Domain**	
Anxiety and Stress	- Generalized Anxiety Disorder 7-item (GAD-7)
Emotional States	- Positive and negative affect scale (PANAS)
Interoceptive Awareness	- Multidimensional Assessment of Interoceptive Awareness (MAIA)
Well-being and Quality of Life	- Ryff’s Psychological Well-Being Scale - Short-Form Health Survey (SF-36)
**Control Variables**	
Sociodemographic and Medical History Information	- Self-reported questionnaire
Sleep Time and Quality	- ActiGraph GT3X Accelerometer
Dietary Habits	- 5-day Dietary Recalls - PREDIMED Questionnaire - Food Frequency Questionnaire (FFQ)

## Data Availability

The raw data supporting this article will be made available by the authors on request.

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
