# Peer review of "Internet-Based Psycho-Physical Exercise Intervention Program in Mild-to-Moderate Depression: The Study Protocol of the SONRIE Randomized Controlled Trial"

_ijerph, 2025, doi:10.3390/ijerph22040540_

Round 1

Reviewer 1 Report

Comments and Suggestions for Authors

I appreciate the opportunity to review this attractive and well-structured randomized clinical trial protocol. I commend the authors for the considerable effort devoted to crafting a manuscript with clear objectives and thoughtful execution. I highlight the great relevance of the topic addressed, especially in the current context, where remote interventions for mental health are of utmost importance. I believe this is a solid and valuable study. Its main strengths lie in its rigorous methodological design, the addressing of a pertinent topic, and the potential impact of its results on clinical practice and future research.

Therefore, I recommend accepting this manuscript, although I suggest that the authors consider the following minimal suggestions to further enrich their work:

  • Provide a stronger rationale for the combination of physical exercise and iCBT in the introduction.
  • Add specific details about the content and duration of the iCBT modules in the methods section.
  • Justify the choice of exercise intensity thresholds based on the Borg RPE scale and ACSM recommendations.

The SONRIE study article appears to be a well-designed clinical trial protocol investigating a promising intervention for mild to moderate depression. I am confident that these small clarifications will further strengthen the impact and clarity of the work.

Comments on the Quality of English Language

The quality of the English language is generally clear and understandable

Author Response

REVIEWER #1

  1. Specific comment

Provide a stronger rationale for the combination of physical exercise and iCBT in the introduction.

Answer

Thank you for your insightful feedback. We have made substantial revisions to the Introduction, which include a complete restructuring and rewriting of this section. As part of these modifications, we have strengthened the rationale for the combination of physical exercise and iCBT, explicitly integrating theoretical and empirical support for their synergistic effects in the treatment of depression. Please, see Introduction section.

  1. Specific comment

Add specific details about the content and duration of the iCBT modules in the methods section.

Answer

Thank you for your suggestion. We have now included specific details regarding the content and duration of the iCBT modules in the Methods section. These details can be found on page 12, lines 433–444.

  1. Specific comment

Justify the choice of exercise intensity thresholds based on the Borg RPE scale and ACSM recommendations.

Answer

Thank you for your comment. We have added information in the Methods section. Specifically, exercise intensity was individually prescribed using the Borg RPE scale to ensure participants trained at moderate-to-vigorous levels, as recommended by the ACSM. The progression from an initial RPE of 3–4 to 7–8 over the 12-week program supports a gradual and safe adaptation. (Page 11, Lines 391 to 393, and 400-414).

Reviewer 2 Report

Comments and Suggestions for Authors

This manuscript presents a study protocol for a randomized controlled trial (RCT) investigating the effectiveness of an internet-based psycho-physical exercise intervention program for mild-to-moderate depression. Here's a critical review, focusing on major and minor points:

Major Comments:

  • The introduction effectively highlights the relevance of the study, particularly in the context of the COVID-19 pandemic. However, the connection between the endocannabinoid system, physical exercise, and depression, while interesting, could be more tightly integrated into the overall rationale. The manuscript needs to show a more clear link between the ES and the intervention. Furthermore, the manuscript should more thoroughly discuss the existing literature regarding the effects of physical activity on depression. Recent studies have demonstrated the positive impacts of combined mindfulness and physical activity interventions on psychological factors and sleep quality in individuals with major depressive disorder (MDD) (Norouzi et al., 2024). Additionally, a systematic review and meta-analysis has shown the positive effect of physical activity on sleep quality in MDD patients (Khazaie et al., 2023). It has also been shown that better sleep quality and higher physical activity levels predict lower emotion dysregulation among persons with MDD (Rezaie et al., 2023). The authors should consider citing and discussing these relevant studies to strengthen the foundation of their rationale and contextualize their intervention within the current body of research.
  • The study's aim is well-defined, but the specific hypotheses should be more explicitly stated. What are the expected outcomes for the experimental group compared to the control group? Particularly, how do the authors expect their physical activity intervention to impact depression, sleep, and emotion regulation, in light of the findings of Norouzi et al. (2024), Khazaie et al. (2023), and Rezaie et al. (2023)?
  • The manuscript mentions that the study started before the COVID-19 pandemic. It would be very helpful to detail how the pandemic impacted the study, and if any changes to the protocol were made.
  • While the manuscript mentions the combination of iCBT and physical exercise, it lacks a detailed description of the intervention components. What specific iCBT modules are included? What type and intensity of physical exercise are prescribed? How is adherence monitored?
  • The manuscript needs to provide more detail about the online platform used for the intervention.
  • The manuscript needs to provide more detail about the qualifications of the people providing the iCBT and the exercise programs.
  • The primary outcome measure (BDI) is appropriate, and the secondary outcome measures are comprehensive. However, the rationale for selecting specific measures should be more explicitly stated.
  • The data collection protocol is well-structured, but it's important to address potential biases and limitations. For example, how is blinding maintained during data collection? How are missing data handled?
  • The manuscript mentions the use of the ActiGraph GT3X accelerometer. It is very important to detail how the data from this device will be processed and analyzed.
  • The manuscript mentions that blood samples will be taken. More detail regarding the handling and processing of these samples needs to be added.
  • The sample size calculation is adequate, but the assumptions underlying the calculation should be clearly justified.
  • The randomization procedure is well-described, but it's important to ensure that the randomization sequence is concealed from the researchers and participants.
  • The dropout rate is a significant concern, particularly in the control group. The manuscript should discuss potential reasons for the high dropout rate and how it might affect the study's findings.
  • The manuscript needs to discuss how the data will be analyzed in light of the high dropout rate.
  • The study's ethical approval is mentioned, but it's important to address other ethical considerations, such as informed consent, data privacy, and potential risks and benefits of the intervention.
  • The manuscript needs to discuss how adverse events will be monitored and reported.

Minor Comments:

  1. Abstract:
    • The abstract is well-written, but it could be more concise and informative.
    • The abstract needs to include the primary outcome measure.
  2. Introduction:
    • The literature review is generally good, but some sections could be more focused.
    • The manuscript could benefit from a clearer definition of "psycho-physical exercise intervention."
  3. Materials and Methods:
    • The flow of participants diagram (Figure 1) is helpful, but it could be more detailed.
    • Table 1 is very helpful.
    • The description of the physical fitness tests is good, but some sections could be more concise.
    • The manuscript uses a lot of abbreviations. It would be helpful to create a table of abbreviations.
  4. Language and Style:
    • The manuscript is generally well-written, but some sentences could be rephrased for clarity and conciseness.
    • Proofread carefully for minor grammatical errors and typos.
  5. ClinicalTrials.gov Identifier:
    • It is very good that the clinical trials identifier is provided.

Author Response

REVIEWER #2

  1. Specific comment

The introduction effectively highlights the relevance of the study, particularly in the context of the COVID-19 pandemic. However, the connection between the endocannabinoid system, physical exercise, and depression, while interesting, could be more tightly integrated into the overall rationale.

Answer

Thank you for this important suggestion. We have revised the introduction to clarify the connection between the endocannabinoid system, physical exercise, and depression. Please, see revised introduction, page 1-2, lines 36-88.

  1. Specific comment

The manuscript needs to show a clearer link between the ES and the intervention.

Answer

Thank you for your comment. In response, we have revised the Introduction to clarify the connection between the ES and our intervention. Please, see revised introduction, page 2, lines 59–74.

  1. Specific comment

Furthermore, the manuscript should more thoroughly discuss the existing literature regarding the effects of physical activity on depression. Recent studies have demonstrated the positive impacts of combined mindfulness and physical activity interventions on psychological factors and sleep quality in individuals with major depressive disorder (MDD) (Norouzi et al., 2024). Additionally, a systematic review and meta-analysis has shown the positive effect of physical activity on sleep quality in MDD patients (Khazaie et al., 2023). It has also been shown that better sleep quality and higher physical activity levels predict lower emotion dysregulation among persons with MDD (Rezaie et al., 2023). The authors should consider citing and discussing these relevant studies to strengthen the foundation of their rationale and contextualize their intervention within the current body of research.

Answer

Thank you for this important recommendation. We have explicitly integrated and discussed recent evidence from Norouzi et al. (2024), Khazaie et al. (2023), and Rezaie et al. (2023) in the revised introduction.  Please, see page 2, lines 46–58.

  1. Specific comment

The study's aim is well-defined, but the specific hypotheses should be more explicitly stated. What are the expected outcomes for the experimental group compared to the control group? Particularly, how do the authors expect their physical activity intervention to impact depression, sleep, and emotion regulation, in light of the findings of Norouzi et al. (2024), Khazaie et al. (2023), and Rezaie et al. (2023)?

Answer

Thank you for this suggestion. We have clarified our hypotheses. Please, see page 3, lines 104–112.

  1. Specific comment

The manuscript mentions that the study started before the COVID-19 pandemic. It would be very helpful to detail how the pandemic impacted the study, and if any changes to the protocol were made.

Answer

Thank you for highlighting this important aspect. As suggested, we have explicitly clarified in the 'Materials and Methods' section (page 3, lines 128–132) how the COVID-19 pandemic impacted our study protocol. Specifically, we indicated that the study was adapted from a face-to-face format to an entirely online intervention precisely at the onset of the COVID-19 confinement period, thus ensuring consistent conditions and adherence for all participants throughout the intervention period.

  1. Specific comment

While the manuscript mentions the combination of iCBT and physical exercise, it lacks a detailed description of the intervention components. What specific iCBT modules are included? What type and intensity of physical exercise are prescribed? How is adherence monitored?

Answer

Thank you for your valuable feedback. We have now provided additional details on both the physical exercise and iCBT components. Please, see Page 10-11, Lines 373-425, and Tables A1 and A2.

  1. Specific comment

The manuscript needs to provide more detail about the online platform used for the intervention.

Answer

Thank you for your comment. Information about the online platform has been added. Please, See page 11, Lines 388-390.

  1. Specific comment

The manuscript needs to provide more detail about the qualifications of the people providing the iCBT and the exercise programs.

Answer

Thank you for your comment. We have now added more detailed information regarding the qualifications of the professionals delivering both the exercise and iCBT interventions. Please see the revised Methods section (Page 11, Lines 384-386 and, Lines 428 to 431).

  1. Specific comment

The primary outcome measure (BDI) is appropriate, and the secondary outcome measures are comprehensive. However, the rationale for selecting specific measures should be more explicitly stated.

Answer

Thank you for your insightful comment. We agree that the rationale for selecting secondary outcome measures should be explicitly stated. To address this, we have now added a justification at the beginning of Section 2.6. (Page 7, Lines 219 to 223).

  1. Specific comment

The data collection protocol is well-structured, but it's important to address potential biases and limitations. For example, how is blinding maintained during data collection? How are missing data handled?

Answer

Comment appreciated. Thank you for highlighting these points. We have added the requested details in the Methods section—specifically in Sections 2.3 (Sample size and randomization), 2.4 (Data Collection), and 2.8 (Statistical Analysis). These additions clarify how potential biases were minimized during data collection and how missing data were handled throughout the study (Page 5, Lines 194 to 198, and Page 13; lines 480 to 487).

  1. Specific comment

The manuscript mentions the use of the ActiGraph GT3X accelerometer. It is very important to detail how the data from this device will be processed and analyzed.

Answer

Thank you for your valuable suggestion. We have now clarified the data processing methodology for accelerometer-derived physical activity and sedentary behavior. You can see changes in section 2.6.1. (Page 8, Lines 240 to 249)

  1. Specific comment

The manuscript mentions that blood samples will be taken. More detail regarding the handling and processing of these samples needs to be added.

Answer

Thank you for your comment. We have incorporated your suggestion by adding additional details on the handling and processing of blood samples in the Methods section, clarifying the collection procedures, storage conditions, and sample processing steps. Please, See Section 2.6.1 (Blood Biomarker Analysis), Page 9, Lines 287 to 327.

  1. Specific comment

The sample size calculation is adequate, but the assumptions underlying the calculation should be clearly justified.

Answer

Thank you for your comment. We have now included a justification for the assumptions used in the sample size calculation. Please, see Page 4 and 5, Lines 152 to 160.

  1. Specific comment

The randomization procedure is well-described, but it's important to ensure that the randomization sequence is concealed from the researchers and participants.

Answer

Thank you for your comment. The concealment of the randomization sequence is addressed in Section 2.3 (Page 5, Lines 168 to 173), where we describe the centralized procedure managed by an independent researcher.

  1. Specific comment

The dropout rate is a significant concern, particularly in the control group. The manuscript should discuss potential reasons for the high dropout rate and how it might affect the study's findings.

The manuscript needs to discuss how the data will be analyzed in light of the high dropout rate.

Answer

Thank you for your comment. We have added a brief explanation in Section 2.3 regarding possible reasons for the higher dropout rate in the control group, and clarified in Section 2.8 how the data will be analyzed to account for attrition. Additionally, we address this issue in the Discussion section by highlighting that dropout is common in internet-based interventions and explaining how we plan to manage its potential impact. Please, see Page 5, Lines 179 to 185, and Page 13, Lines 505 to 509.

  1. Specific comment

The study's ethical approval is mentioned, but it's important to address other ethical considerations, such as informed consent, data privacy, and potential risks and benefits of the intervention.

Answer

Thank you for your comment. We have now included additional details regarding informed consent, data privacy, and the potential risks and benefits of the intervention. These modifications can be found on page 3, lines 118–125 of the revised manuscript.

  1. Specific comment

The manuscript needs to discuss how adverse events will be monitored and reported.

Answer

Thank you for this valuable observation. We have added a detailed description in the Methods section (Intervention) explaining how participant safety was monitored throughout both components of the intervention. Please, see Page 11, Lines 415 to 423, and Page 12, Lines 444 to 453.

  1. Specific comment

The abstract is well-written, but it could be more concise and informative.

The abstract needs to include the primary outcome measure.

Answer

Thank you. We have revised the abstract to improve conciseness and clarity. In the updated version, we have explicitly stated that the primary outcome is the change in depression severity, as measured by the Beck Depression Inventory.

  1. Specific comment

The literature review is generally good, but some sections could be more focused.

The manuscript could benefit from a clearer definition of "psycho-physical exercise intervention."

Answer

Thank you for your helpful feedback. In response, we reviewed the Introduction to improve clarity and focus, ensuring that each section directly supports the rationale for the intervention. Additionally, we added a clear definition of the term “psycho-physical exercise intervention” at the beginning of Section 2.7.2. Please, See Page 10, Lines 373 to 382.

  1. Specific comment

The flow of participants diagram (Figure 1) is helpful, but it could be more detailed.

Answer

We have revised Figure 1 to include additional details regarding participant flow, such as reasons for exclusion, specific causes of dropout at each stage (e.g., personal reasons, COVID-19 illness, lost contact), and timing of recruitment and Intervention. Please, see Figure 1, Page 4, Lines 139- 140.

  1. Specific comment

Table 1 is very helpful.

Answer

Thank you.

  1. Specific comment

The description of the physical fitness tests is good, but some sections could be more concise.

Answer

Thank you for your suggestion. We have carefully revised the description of the physical fitness assessments to improve clarity and conciseness, eliminating redundant phrasing while retaining all relevant methodological details. Please see Page 8, Lines 252 to 285.

  1. Specific comment

The manuscript uses a lot of abbreviations. It would be helpful to create a table of abbreviations.

Answer

Done

  1. Specific comment

The manuscript is generally well-written, but some sentences could be rephrased for clarity and conciseness.

Proofread carefully for minor grammatical errors and typos.

Answer

Thank you for your feedback. We have carefully proofread the manuscript and made necessary revisions to improve clarity, conciseness, and correct minor grammatical errors and typos. We have aimed to ensure that the text is as clear and precise as possible.

  1. Specific comment

It is very good that the clinical trials identifier is provided.

Answer

Thank you for your positive feedback.

Round 2

Reviewer 2 Report

Comments and Suggestions for Authors

The authors now have addressed my concerns, I would like to accept this manuscript in this version,